# Synthesis of Novel Nanocomposite Materials with Enhanced Antimicrobial Activity based on Poly(Ethylene Glycol Methacrylate)s with Ag, TiO_2_ or ZnO Nanoparticles

**DOI:** 10.3390/nano14030291

**Published:** 2024-01-31

**Authors:** Melpomeni Tsakiridou, Ioannis Tsagkalias, Rigini M. Papi, Dimitris S. Achilias

**Affiliations:** 1Laboratory of Polymer and Colors Chemistry and Technology, Department of Chemistry, Aristotle University of Thessaloniki, 54124 Thessaloniki, Greece; melptsak@chem.auth.gr (M.T.); itsagkal08@gmail.com (I.T.); 2Laboratory of Biochemistry, Department of Chemistry, Aristotle University of Thessaloniki, 54124 Thessaloniki, Greece; rigini@chem.auth.gr

**Keywords:** nanocomposites, oligo(ethylene glycol methacrylate), Ag NPs, TiO_2_, ZnO, antimicrobial activity

## Abstract

The aim of this investigation was to prepare novel hybrid materials with enhanced antimicrobial properties to be used in food preservation and packaging applications. Therefore, nanocomposite materials were synthesized based on two stimuli-responsive oligo(ethylene glycol methacrylate)s, namely PEGMA and PEGMEMA, the first bearing hydroxyl side groups with three different metal nanoparticles, i.e., Ag, TiO_2_ and ZnO. The in situ radical polymerization technique was employed to ensure good dispersion of the nanoparticles in the polymer matrix. FTIR spectra identified the successful preparation of the corresponding polymers and XRD scans revealed the presence of the nanoparticles in the polymer matrix. In the polymer bearing hydroxyl groups, the presence of Ag-NPs led to slightly lower thermal stability as measured by TGA, whereas both ZnO and TiO_2_ led to nanomaterials with better thermal stability. The antimicrobial activity of all materials was determined against the Gram-negative bacteria *E. coli* and the Gram-positive *S. aureus*, *B. subtilis* and *B. cereus*. PEGMEMA nanocomposites had much better antimicrobial activity compared to PEGMA. Ag NPs exhibited the best inhibition of microbial growth in both polymers with all four bacteria. Nanocomposites with TiO_2_ showed a very good inhibition percentage when used in PEGMEMA-based materials, while in PEGMA material, high antimicrobial activity was observed only against *E. coli* and *B. subtilis*, with moderate activity against *B. cereus* and almost absent activity against *S. aureus*. The presence of ZnO showed antimicrobial activity only in the case of PEGMEMA-based materials. Differences observed in the antibacterial activity of the polymers with the different nanoparticles could be attributed to the different structure of the polymers and possibly the more efficient release of the NPs.

## 1. Introduction

Stimuli-responsive polymers are among the most promising classes of modern polymeric materials; their main characteristic is that they alter their physical properties in response to external stimuli, and as such, they have grown significantly over the last two decades [1]. They can respond to changes in their environment such as temperature, pH and the presence of electrolytes and show significant changes in volume due to small changes in pH, temperature, electric field and light. They can exhibit positive swelling thermal sensitivity, in which polymers with an upper critical solution temperature (UCST) shrink on cooling below the UCST. Among other materials, (meth)acrylates bearing short oligo(ethylene glycol) (OEG) side parts have been exponentially studied during the last few years [2,3,4]. These macro-monomers are polymerized to produce poly(meth)acrylates, which are functionalized with various-length OEG units. These thermoresponsive and biocompatible materials can be used for a variety of purposes, including drug release capsules, injectable hydrogels, biosensors, smart gels for chromatography, smart food packaging, antimicrobial surfaces and artificial tissues [5,6,7].

Recently, in our group we studied the radical polymerization kinetics of two commercially available oligo(ethylene glycol methacrylates) [8], namely oligo(ethylene glycol) methyl ether methacrylate with 4–5 ethylene oxide units (OEGMMA, M_n_ = 300, x = 4–5), which after polymerization results in the polymer given the name PEGMEMA, and oligo(ethylene glycol) hydroxyethyl methacrylate, with nearly x = 6 ethylene oxide units resulting in the PEGMA polymer (Figure 1). It was found that side ethylene glycol and hydroxyl groups affect polymerization kinetics with PEGMA, exhibiting faster reaction kinetics.

When in direct contact with water, both PEGMA and PEGMEMA adsorb a significant percentage of water by weight of material (the percentage can vary from 60–85%) and swell, and the final product transforms into a hydrogel. Hydrogels are materials which are not water-soluble but have the property of binding significant amounts of water inside them [9,10,11]. Hydrogels based on PEGMA or PEGMEMA are popular in the biomedical and pharmaceutical fields due to their increased chemical stability and excellent biocompatibility [12,13]. 

Nanocomposite materials based on a polymeric matrix have attracted considerable interest in both research and industry due to their improved properties as well as low production costs [14]. The addition of small amounts of nanoadditives in the polymer matrix can lead to new materials with superior thermal, mechanical and gas barrier properties without hindering their biodegradable and non-toxic characters [15,16,17,18]. The main processes for the preparation of polymer-based nanocomposite materials include (a) solution casting, (b) melt blending and (c) the in situ polymerization [19]. In the first two, the polymer is used, whereas in the third, the monomer, together with the nanomaterial and the polymeric nanocomposite, is formed after polymerization. The compatibility between nanomaterials and the polymer matrix is considered a challenge for the preparation of the nanocomposites. In order to obtain a good dispersion of the nanoparticles in the polymer matrix, the in situ polymerization was followed here. 

Wide varieties of nanomaterials are suitable for offering smart and/or intelligent properties for food packaging materials, as demonstrated by oxygen scavenging capability or antimicrobial activity. The most favorable nanomaterials are layered silicate nanoclays, for example montmorillonite (MMT) and kaolinite, zinc oxide (ZnO), titanium dioxide (TiO_2_) and silver nanoparticles (Ag-NPs) [20,21,22,23]. The last three nanoparticles were employed in this investigation in order to examine the antimocribal activity which they can provide to the nanocomposite materials. The high performance of NPs in nanocomposite antimicrobial systems is mostly successful due to the high surface-to-volume ratio and elevated surface reactivity of the nano-sized antimicrobial metal/metal oxide particles enabling them to inactivate microorganisms more efficiently than their micro- or macro-scale equivalent [21].

Silver nanoparticles (AgNPs) have been known for about 120 years, whereas silver has been used since ancient times as a means of healing burns and chronic wounds. In recent years, the use of silver as an antimicrobial agent has been thoroughly studied due to the emergence of antibiotic-resistant bacteria [24,25,26,27,28]. Usually, AgNPs are synthesized by reducing silver salts (such as AgNO_3_) in suitable reducing agents such as ethylene glycol, organic solvents or water. Silver exhibits antimicrobial properties due to its ability to react with the sulfur compounds found in the respiratory system of bacterial cells; it adheres to the cell walls and cell membranes of the bacteria and inhibits the respiratory function. Silver nanoparticles show an enhanced antimicrobial character compared to larger silver particles due to the large specific surface area, which offers better contact with microorganisms. The mechanism for dealing with microorganisms remains the same, but due to the size and number of nanoparticles, it becomes easier to adhere to the cell membrane as well as to penetrate inside the microorganisms [29,30,31]. In food packaging, nanosized particles of silver inhibit the growth of microorganisms due to their broad-spectrum inhibitory activities. Furthermore, exhibiting antibacterial activity for prolonging shelf life, nano-Ag could catalyze the absorption and decomposition of ethylene emitted from fruit metabolism, which has been considered as an ethylene blocker [21].

Metal oxides (such as TiO_2_, ZnO), have attracted considerable attention for packaging applications because of their unique chemical and physical properties such as antibacterial activity, thermal stability and low toxicity [21]. The incorporation of ZnO NPs in the polymer of a food packaging material allows interactions between the packaging and the food and thus has a vigorous effect on its preservation [32]. ZnO nanoparticles are also versatile systems for biomedical applications. More specifically, there have been therapeutic interventions and recent studies that prove that they are promising anti-cancer systems. Through its mechanisms of action, the cellular consequences resulting from the interactions of nanoparticles with cells, the inherent toxicity and anti-cancer selectivity of ZnO can be further improved as a new attractive anticancer system [33,34].

Due to its excellent whiteness and high refractive index, TiO_2_ is the dominant white pigment for paints, paper, plastics, rubber, toothpaste, food and various other materials. Its properties, such as the high optical refractive index and transparency in the visible spectrum and superhydrophilicity, enable a range of practical applications. Recent applications of TiO_2_ nanoparticles include the cosmetics industry and glass ceramics together with the food and bakery industries. When incorporated into food packaging films, TiO_2_ NPs could prevent the food component from oxidation by UV irradiation while retaining good transparency due to its UV blocking property [35]. In food processing, the biocide effect of TiO_2_ nanocomposites with polymer materials, such as ethylene vinyl alcohol and chitosan, has been studied [36]. TiO_2_ showed the highest antibacterial effect on Gram-positive rather than Gram-negative microorganisms [37].

Although the properties of polymeric hydrogels based on oligo(ethylene glycol) methacrylates have been studied in the literature, according to our knowledge, no studies have been published on their use as a matrix to form nanocomposite materials. Moreover, these stimuli-responsive materials could potentially be an alternative to synthetic plastic packaging, and their combination with nanofillers enables the development of advanced smart food packaging materials with enhanced properties. Therefore, in this investigation, the formation of nanocomposites based on PEGMA and PEGMEMA was investigated with Ag-NPs, ZnO or TiO_2_ prepared via in situ polymerization. The chemical structure and thermal stability of the materials was measured via ATR-FTIR, XRD and TGA, and focus was given to their antimicrobial activity against both Gram-positive and Gram-negative bacteria.

## 2. Materials and Methods

### 2.1. Materials

The monomers used were oligo(ethylene glycol) methyl ether methacrylate with 4–5 ethylene oxide units, (OEGMMA, Mn = 300, d = 1.05 g/mL, purity > 98%) and oligo(ethylene glycol) hydroxyl ethyl methacrylate with nearly 6 ethylene oxide units (OEGHEMA, Mn = 360, d = 1.105 g/mL, purity > 97%), both from Sigma-Aldrich (St. Louis, MI, USA). The free radical initiator used in the polymerization experiments was benzoyl peroxide (BPO) (purity > 97 %, from Fluka, Buchs, Switzerland), and it was purified by fractional recrystallization twice from methanol (Merck, Rahway, NJ, USA). AgNO_3_ used as a precursor for the formation of AgNPs was supplied by Mallinckrodt (C = 0.05 mol/L). ZnO was purchased from Aldrich in the form of nanopowder with a size < 100 nm (d = 5.61 g/mL). TiO_2_ was also supplied by Aldrich in the form of nanopowder with a size < 100 nm (d = 3.8 g/mL). All other chemicals used were of reagent grade.

### 2.2. Preparation of the Neat Polymers via Radical Polymerization

The synthesis of PEGMA and PEGMEMA neat polymers began with the addition of 4 mL of the macromonomers, i.e., either OEGHEMA or OEGMMA, in the polymerization vessel. Then, 0.0291 g of the initiator, BPO, was added corresponding to the concentration of the initiator in the monomer equal to 0.03 M. Next, the mixture was placed in an ultrasonic bath (Xuba3, Grant, Cambridge, UK), where it remained for a period of about 5 min; this time is sufficient for the dissolution of the initiator in the macromonomer. Following this, Nitrogen (N_2_) gas was introduced into the polymerization vessels in order to form an inert atmosphere and remove oxygen, which acts as an inhibitor during the polymerization reaction. The reaction vessels were sealed and placed inside a thermostated water bath, which was at a temperature of 80 °C. The stirring frequency of the shaker was adjusted to 120 rpm. The polymerization process takes about 30 min, the time necessary for the full polymerization of the monomers.

### 2.3. Preparation of the Nanocomposite Materials via In Situ Radical Polymerization 

The synthesis process of the nanocomposite polymers with each of the nanoadditives, i.e., AgNO_3_, ZnO or TiO_2,_ is identical to that of the neat polymers, with the only difference being the addition of additives. Six nanocomposites were prepared including either PEGMA or PEGMEMA as the matrix polymer with each one of the nanoadditives, i.e., AgNO_3_, ZnO, TiO_2_. Then, 2 mL of each of the macromonomers was added into the polymerization vessel and weighed. Then, the appropriate amounts of AgNO_3_, ZnO or TiO_2_ were introduced. The amount of each nanoadditive was 0.01105 g in the case of PEGMA and 0.0105 g in the case of PEGMEMA. Then, the mixtures were placed in the ultrasonic bath for approximately 7 min to form a uniform distribution of the additives in the mixture. Next, the necessary amount of BPO was weighed to meet the concentration requirement of 0.03 M BPO in PEGMA and PEGMEMA, and it was added to the mixture. Inert atmosphere was provided by the addition of Nitrogen in the vessels. The vessels were placed in the water bath already set at the desired polymerization temperature of 80 °C. Polymerization took place for 30 min. After the reaction, all isolated nanocomposite materials were dried to a constant weight in a vacuum oven at room temperature and stored for the measurements. It should be noted here that the cations Ag^+^ originally formed in the mixture from the added AgNO_3_ solution were reduced to Ag NPs during polymerization due to the presence of hydroxyl and carbonyl groups in the monomer and polymer molecules. 

### 2.4. Analytical Techniques

*Fourier-Transform Infra-Red (FTIR) Spectroscopy.* The chemical structure of the neat polymers and the nanocomposite materials was confirmed by recording their IR spectra. The instrument used was the Spectrum One spectrophotometer from Perkin Elmer (Akron, OH, USA) with an attenuated total reflectance (ATR) device. IR spectra were recorded over a range from 4000 to 700 cm^−1^ at a resolution of 2 cm^−1^, and 32 scans were averaged to reduce noise. IR radiation first hit a zinc selenide (ZnSe) plate and then found the sample at an angle of 45°. The instrument’s software (Spectrum v5.0.1) was used to identify several peaks.

*Wide Angle X-Ray Diffraction Patterns (WAXRD).* X-ray diffraction measurements of the samples were performed using a MiniFlex II XRD diffractometer from Rigaku Co., Ltd. (Chalgrove, Oxford, UK), with CuKa radiation at wavelength λ = 0.1540 nm. The angular range over which the measurements were made was 2θ = 5–85° and the scanning speed was 2° per minute.

*Thermogravimetric Analysis (TGA).* TGA analysis was performed on a Pyris 1 TGA (Perkin-Elmer, Akron, OH, USA) thermal analyzer. In a typical temperature scan experiment, 3–4 mg of sample were heated from ambient temperature to 600 °C at a heating rate of 20 °C/min under nitrogen flow.

### 2.5. Determination of Antimicrobial Properties

The antibacterial activity of the synthesized nanocomposite materials was evaluated against four different microbial species, namely *Escherichia coli* (BL21), *Staphylococcus aureus* (ATCC 25923), *Bacillus subtilis* (ATCC 6633) and *Bacillus cereus* (ATCC 11778), by the disc diffusion method for the antimicrobial susceptibility test [32,33,34].

The Luria-Bertani (LB) broth used included 1% *w*/*v* tryptone (AppliChem), 0.5% *w*/*v* yeast extract (AppliChem, GmbH, Darmstadt, Germany) and 0.5% *w*/*v* NaCl (Merck, KGaA, Darmstadt, Germany). The Agar used was from Honeywell Fluka. To prepare the nutrient culture medium LB broth, tryptone, yeast extract and NaCl were mixed with ddH_2_O. In case it was desired to create a solid nutrient medium for plating petri-dishes, additional agar was added at a ratio of 2% *w*/*v*. This was followed by immediate sterilization of the nutrient materials in a liquid sterilization oven (autoclave) at a temperature of 121 °C and pressure of 1.3 bar for a period of 20 min. For the solid bacterial cultures, 15 ml of fluid LB-agar medium per petri dish (100 mm × 15 mm, Corning Costar, Cambridge, MA, USA) was used.

Bacteria were grown in 10 mL LB sterile medium at 150 rpm and 37 °C. They were then harvested at the logarithmic growth stage and the concentration of the suspensions was adjusted to an OD600 (optical density at 600 nm) value of 0.6 in 25 mM buffer (PBS, Sigma) prior to incubation with each material. More specifically, 500 µL of each bacterial culture was incubated with 100 mg of each nanoparticle in sterile 3.5 cm diameter dishes at 37 °C, with gentle agitation for 1 h. The culture fluid was removed, the material was washed twice in succession and the remaining cells were picked up with 1 ml of LB. Serial 10-fold dilutions of the cells obtained with LB and 100 µL were carried out in Eppendorf. From each dilution, they were plated in 10 cm diameter petri dishes containing LB agar. The plates were incubated at 37 °C for 24 h and the resulting colonies were counted to calculate the number of colony-forming units per mL. PEGMA and PEGMEMA free of silver, titanium and zinc were used as negative controls.

## 3. Results and Discussion

### 3.1. Characterization of the Samples

FTIR spectra of neat PEGMA and their nanocomposites appear in Figure 1. Regarding the neat PEGMA polymer, the stretching vibration of the carbonyl (C=O) bond is identified as a strong and sharp peak in the region of about 1724.5 cm^−1^. The stretching vibration of the hydroxyl group (O-H) is located as a broad peak in the region of about 3456 cm^−1^; the stretching vibrations of the C-H bonds (aliphatic groups CH_2_) are located in the region of 2867cm^−1^. The sharp peak observed at 1451cm^−1^ is due to the bending vibration of the -CH_2_- group and the distortion of the -CH_3_ group. The broad band at 1092 cm^−1^ is ascribed to the C-O-C group (characteristic of the specific polymer). Finally, the peak at 1246 cm^−1^ is related to strain vibrations of the ester groups. All the above-mentioned peaks are found in spectra for PEGMA in the literature [38].

Regarding the PEGMA nanocomposites with Ag, TiO_2_ and ZnO nanoparticles, a decrease in the intensity of the peak attributed to the carbonyl group is observed. Specifically, from the value at 1724.5 cm^−1^ in the free polymer, the peak is located at 1722 cm^−1^ in the case of Ag, at 1716 cm^−1^ in the case of TiO_2_ and at 1720 cm^−1^ in the case of ZnO. This observation reveals the interaction of the carbonyl group with Ag, TiO_2_ and ZnO. It is worth mentioning the absence of any shift of the peak attributed to the O-H group (3456 cm^−1^), which implies the absence of interaction of this group with Ag, TiO_2_ and ZnO.

FTIR spectra of neat PEGMEMA and their nanocomposites appear in Figure 2. Regarding the neat polymer, the carbonyl bond (C=O) strain vibration is identified as a strong and sharp peak in the region of approximately 1725.5 cm^−1^. In the region of 2870 cm^−1^, the stress vibrations of the C-H bonds (aliphatic CH_2_ groups) are found. The sharp peak observed at 1451 cm^−1^ is due to the bending vibration of the -CH_2_- group and the distortion of the -CH_3_ group. The broad band at 1098 cm^−1^ is assigned to the C-O-C functional group. Finally, the peak at 1246 cm^−1^ is related to strain vibrations of the ester groups. All the above-mentioned peaks are found in spectra for PEGMEMA in the literature [39]. Regarding the PEGMEMA nanocomposites with Ag, TiO_2_ and ZnO nanoparticles, no decrease in the intensity of the peak attributed to the carbonyl group is observed. The same is observed for the remaining peaks of the FTIR spectra.

The crystalline nature of the synthesized nanocomposites was studied by means of X-ray diffractometry. The diffraction patterns for the polymers as well as for the synthesized nanocomposites are shown in Figure 3a and Figure 3b for the PEGMA and PEGMEMA nanocomposites, respectively.

The broad peaks appearing in all samples in Figure 3a at 21.5° and 42° are related to the amorphous nature of the PEGMA polymer. With the addition of Ag, TiO_2_ and ZnO, the broad peaks recorded in the neat polymer also appear in the diffractograms of the corresponding materials. However, the intensity of the peaks gradually decreases in the materials containing Ag, ZnO and TiO_2_, respectively, which indicates the interaction of the compounds thereof with the free PEGMA polymer.

The broad peaks appearing in all samples in Figure 3b at 21.5° and 43° are related to the amorphous nature of the PEGMEMA polymer. With the addition of Ag, TiO_2_ and ZnO, the broad peaks recorded in the free polymer also appear in the diffraction patterns of the corresponding materials. However, the intensity of the peaks gradually increases in order in the materials containing Ag, TiO_2_ and ZnO (in these two, the intensity of peaks is almost identical), which indicates the interaction of these compounds with the free PEGMEMA polymer. For silver, the peaks observed at 38.2° and 44.4° are assigned to the (111), (200) crystallographic planes of the face-centered cubic silver [28]. For titanium oxide, the peaks at 40.2°, 43.5° reveal the presence of TiO_2_ in the rutile phase, while small peaks at 25.1°, 47.2° are attributed to the crystallographic planes (200) and (204) of the phase ascension. For zinc oxide, the characteristic peaks at 32.2°, 34.9°, 36.7°, 48.0°, 57.0° reveal the presence of hexagonal ZnO nanoparticles in the polymer matrix [40]. The intensity of the peaks is low due to the very small amount of additive used, i.e., 1%.

### 3.2. Thermal Stability of the Nanocomposites

The thermal stability of the prepared samples was studied through thermogravimetric analysis. The mass reduction rate results of the PEGMA polymer and its derived materials with Ag, TiO_2_ and ZnO are shown in Figure 4.

From the mass loss plot in Figure 4a, it is evident that the PEGMA materials containing ZnO and TiO_2_ show slightly improved thermal stability compared to the PEGMA polymer and begin to decompose at higher temperatures. This is in accordance with the literature findings [23]. The presence of ZnO or TiO_2_ forms protective barriers contributing to the better thermal stability of the polymer. In addition, Ag-NPs containing PEGMA material are observed to have lower thermal stability compared to the PEGMA polymer (as well as ZnO and TiO_2_-containing PEGMA materials) and begin to decompose at lower temperatures. The peak appearing in the temperature range from 200 °C to 300 °C is related to the destruction of weak bonds present in the macromolecular chain of PEGMA and the materials derived from it with Ag-NPs. Probably, the association of Ag-NPs in the macromolecular chains formed from the reduction of Ag^+^ may lead to a destruction of some weak bonds (such as C=O…HO) originally present in the polymer. This is in accordance with FTIR findings, where some interaction of the carbonyl group with Ag was observed.

The results of the mass reduction rate of the PEGMEMA polymer and the materials derived from it with Ag, TiO_2_ and ZnO are shown in Figure 5.

From the mass loss plot in Figure 5, it is evident that the PEGMEMA materials containing ZnO show higher thermal stability compared to the neat PEGMEMA polymer and start to decompose at higher temperatures. This again is attributed to a thermal barrier formed by this metal oxide protecting the scission of macromolecular chains. PEGMEMA does not include hydroxyl groups, so hydrogen bonds are not formed and the thermal stability of the nanocomposite with Ag-NPs seems similar to that of the neat polymer. In addition, the PEGMEMA material containing TiO_2_ is observed to have a bimodal distribution, denoting two steps in thermal decomposition. This could be attributed to the existence of two populations of macromolecules formed during polymerization: one with a rather lower length in the macromolecular chains responsible for the production of polymer exhibiting initially lower thermal stability compared to neat PEGMEMA and another with a higher macromolecular chain length exhibiting a second peak denoting degradation at temperatures similar to those of neat PEGMEMA.

### 3.3. Determination of In Vitro Antimicrobial Activity 

#### 3.3.1. Cultivation of *E. coli* on PEGMA and PEGMA/NPs

As observed in Figure 6, the PEGMA homopolymer has abundant growth of *E. coli* bacteria, as was expected (Figure 6a). From the three nanoparticles added in PEGMA, i.e., Ag-NPs, TiO_2_ and ZnO, it is found that PEGMA + Ag NPs (Figure 6b) shows very strong antimicrobial activity. In the nanocomposites PEGMA + TiO_2_/NPs and PEGMA + ZnO/NPs, shown in Figure 6c and 6d, respectively, mild and a strong microbial growth were observed, respectively. Note that a 10^−4^ dilution of bacteria cell culture was used for all samples tested.

#### 3.3.2. Cultivation of *E. coli* on PEGMEMA and PEGMEMA/NPs

In Figure 7, the culture of *E. coli* with both the PEGMEMA homopolymer and its composites is presented. Again, the culture was prepared with samples that had been diluted to 10^−4^. It was clear that although in neat PEGMEMA a large growth of *E. coli* bacteria was present, as was expected, in all nanocomposites of PEGMEMA with Ag NPs, TiO_2_ or ZnO strong antimicrobial activity was recorded, as there was no bacterial colony in all plates. Therefore, it seems that PEGMEMA nanocomposites show a better antimicrobial activity compared to those based on PEGMA. This could be attributed to the different structure of the polymers and possibly the more efficient release of the NPs.

#### 3.3.3. Cultivation of *Staphylococcus aureus* on PEGMA and PEGMA/NPs

In Figure 8, it can be observed that in neat PEGMA a mild growth of staphylococcal bacteria exists, while in the nanocomposite of PEGMA with Ag NPs there is no bacterial colony, which indicates the strong antimicrobial effect of Ag nanoparticles against *Staphylococcus aureus*. Conversely, this activity was not observed in the nanocomposites of PEGMA with either TiO_2_ or ZnO NPs. The composite polymers with TiO_2_ or ZnO inclusions have a similar number of colonies to the metal-free PEGMA polymer, which means that the existence of these nanoparticles has a negligible impact on the specific bacterium; therefore, no antimicrobial activity is observed in this composite compared to the neat polymer.

#### 3.3.4. Cultivation of Staphylococcus Aureus on PEGMEMA and PEGMEMA/NPs

In Figure 9, the neat polymer, PEGMEMA, exhibits a minimal growth of bacterial colonies of *Staphylococcus aureus*. In all nanocomposite materials containing Ag, TiO_2_ and ZnO nanoparticles, there is no bacteria colony, so it can be confidently stated that the specific composite polymers with the above metals all have antimicrobial activity against *St. aureus* bacteria. It should be noted that all samples were cultivated with *St. aureus* cell dilutions of the order of 10^−4^.

#### 3.3.5. Cultivation of *Bacillus subtilis* on PEGMA and PEGMA/NPs

In Figure 10, concerning the cultivation of *B. subtilis*, it is illustrated that in neat PEGMA a large number of bacteria have grown as expected. The composites with Ag and TiO_2_ NPs show strong antimicrobial activity as no colonies are observed, in contrast to the composite with ZnO, where a small number of colonies are presented, although not as many as in the neat polymer, which is considered as the negative control standard.

#### 3.3.6. Cultivation of *Bacillus subtilis* on PEGMEMA and PEGMEMA/NPs

In Figure 11 the situation is clear, since only the neat PEGMEMA polymer (negative control standard) shows a large number of *B. subtilis* colonies. All other nanocomposite polymers of PEGMEMA with Ag, TiO_2_ and ZnO NPs have antimicrobial activity against the specific bacterium.

#### 3.3.7. Cultivation of *Baillus cereus* on PEGMA and PEGMA/NPs

The samples of PEGMA and PEGMA/NPs cultivated with the bacterium *B. cereus* appear in Figure 12. It is clear that antimicrobial activity shows only the nanocomposite of PEGMA with Ag NPs, which has no colony at all. All other samples, i.e., PEGMA+TiO_2_/NPs and PEGMA+ZnO/NPs, have several colonies of *Bacillus cereus* but fewer than neat PEGMA. A 10^−4^ dilution of bacteria cell culture was used for all samples tested.

#### 3.3.8. Cultivation of *Bacillus cereus* on PEGMEMA and PEGMEMA/NPs

A similar reaction to PEGMA/NPs was also shown by PEGMEMA/NPs, as can be seen from the image shown in Figure 13, since only the nanocomposite with Ag presents an antimicrobial effect (zero number of colonies). In contrast, neat PEGMEMA and the nanocomposites of PEGMEMA with TiO_2_ and ZnO NPs showed a significant number of *Bacillus cereus* colonies grown. 

#### 3.3.9. Comparative Antimicrobial Results

The following Table 1 summarizes the antimicrobial effect (colonies) of all the materials studied, and the % inhibition of microbial growth is illustrated in Table 2.

It is clear that neat polymers exhibit a large number of colonies of all bacteria except for *S. aureus*. Numerous colonies were also observed in the nanocomposite of PEGMA and PEGMEMA with ZnO and TiO_2_ against *B. cereus*, except the nanocomposite with Ag NPs. In all other materials, a limited number of colonies were observed. 

From Table 2, it is seen that the nanocomposite of PEGMA with Ag NPs exhibits the best inhibition of microbial growth, with three bacteria out of four (*E. coli*, *S. aureus* and *B. subtilis*) showing inhibition action over 90%. In the literature, silver nanoparticles have also shown effective antibacterial activity and other polymer systems [41]. Possible mechanisms of Ag NP action have been proposed in the literature [27,28]. In detail, during incubation, Ag NPs released from the composites gradually diffused in the seeded agar and attached on the bacteria cell wall, disrupting metabolic activities. Adhesion of Ag NPs at the surface causes deformation of the membrane that leads to an increase in the membrane permeability. Ag NPs diffuse into the bacterial cell, causing DNA damage [27,28]. 

Furthermore, the antimicrobial activity of TiO_2_ was high against *E. coli* and *B. subtilis*, moderate against *B. cereus* and almost absent against *S. aureus*. The effect on *E. coli* was systematically stronger than on *S. aureus* [42]. The presence of ZnO did not show a significant antimicrobial activity against any bacteria used in PEGMA.

In the case of PEGMEMA and PEGMEMA nanocomposites, much better antimicrobial activity in all bacteria was observed. Specifically, all nanocomposites with three out of the four bacteria studied, i.e., *E. coli*, *S. aureus* and *B. subtilis*, showed a 100% inhibition of microbial growth. In the case of *B. cereus*, good inhibition was also observed. In conclusion, the antimicrobial activity of PEGMEMA nanocomposites is undoubtedly much greater than that of PEGMA.

A difference between the antibacterial activity of PEGMA polymers with NPs and PEGMEMA with NPs was observed in our experiments, with PEGMEMA presenting a better activity. This could be attributed to the different structure of the polymers and possibly the more efficient release of the NPs.

## 4. Conclusions

In this investigation, nanocomposite materials were prepared based on two polymers having stimuli-responsive characteristics and three different metal nanoparticles, including Ag, TiO_2_ and ZnO. Specifically, the polymers were based on two oligo(ethylene glycol methacrylates), with one also bearing hydroxyl side groups. The in situ radical polymerization method was used to prepare homogeneous nanocomposite materials.

FTIR spectra identified the successful preparation of the corresponding polymers and XRD scans revealed the presence of the nanoparticles in the polymer matrix. In the polymer bearing hydroxyl groups, the presence of Ag-NPs led to lower thermal stability, whereas both ZnO and TiO_2_ led to nanomaterials with better thermal stability. In the case of the polymer without hydroxyl groups, the nanocomposite with ZnO again resulted in material with higher thermal stability, whereas the presence of TiO_2_ resulted in non-uniform macromolecular chains with a dual degradation step and the existence of Ag-NPs in a thermal stability similar to neat polymer.

The antimicrobial activity of all the materials prepared was determined against several bacteria, including the Gram-negative *Escherichia coli* (BL21), the Gram-positive *Staphylococcus aureus* (ATCC 25923), *Bacillus subtilis* (ATCC 6633) and *Bacillus cereus* (ATCC 11778). In these four bacteria, the PEGMEMA nanocomposites had much better antimicrobial activity than the PEGMA composites. Ag NPs exhibited the best inhibition of microbial growth in both polymers with all four bacteria. Nanocomposites with TiO_2_ showed a very good inhibition percentage when used in PEGMEMA-based materials, while in PEGMA material, high antimicrobial activity was observed only against *E. coli* and *B. subtilis*, with moderate activity against B. cereus and almost absent activity against *S. aureus*. The presence of ZnO showed antimicrobial activity only in the case of PEGMEMA-based materials.

## Data Availability

Data are contained within the article.

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
