# Peer review of "Synthesis of Novel Nanocomposite Materials with Enhanced Antimicrobial Activity based on Poly(Ethylene Glycol Methacrylate)s with Ag, TiO2 or ZnO Nanoparticles"

_nanomaterials, 2024, doi:10.3390/nano14030291_

Round 1

Reviewer 1 Report

Comments and Suggestions for Authors Nanomaterials Manuscript Number:  nanomaterials-2831177 Title: "Synthesis, characterization and antimicrobial activity of nanocomposite materials based on poly(ethylene glycol methacrylate)s with silver, TiO2 or ZnO nanoparticles" Author(s): Melpomeni Tsakiridou, Ioannis Tsagkalias, Rigini M. Papi, Dimitris S. Achilias                         In this study, nanocomposite materials were synthesized using two oligo(ethylene glycol methacrylate)s, exhibiting stimuli-responsive characteristics. Three different metal nanoparticles, namely Ag, TiO2, and ZnO, were introduced using in situ radical polymerization. FTIR spectra confirmed the successful preparation of the corresponding polymers, and XRD scans indicated the presence of nanoparticles in the polymer matrix. Antimicrobial activity against Gram-negative bacteria E. coli and Gram-positive bacteria S. aureus, B. subtilis, and B. cereus was evaluated for all materials.   The scientific subject could be of interest from practical point of view, yet, the format and the structure of the manuscript is not acceptable.

Please make sure that the title of your manuscript is carefully crafted to optimize visibility and enhance the publication's overall impact. In this form it is too general.   Page 1, line 6. Laboratory not Lab.   The abstract is not a convincing one; it is too short, slightly minded caching… Also, does not quite draw attention of the reader. I think the first sentence (page 1, lines 11-13) should be deleted. Page 1, line 25, where does the abbreviation PEGMEMA come from?   I did not find anywhere how Ag NPs were obtained. On page 4, line 173 it says that AgNO3 was used.   Page 5, line 227. Figure 1a-1d?   Page 6, line 230 “....is located as a strong and broad peak...” But from figure 1 this is not observed.   Page 6, line 235. Please add bibliographic references here.   Page 6, line 253. Please add bibliographic references here.   Page 8, line 280.” For silver the peaks observed at 38.2o and 44.4o...” From your figure I do not notice the 2 peaks.   Page 9, line 297. “...containing ZnO and TiO2 show greater thermal stability” I think that is too much said. Perhaps it is better to add that it has a slightly improved thermal stability compared to the one observed for the PEGMA graded sample. Page 9, lines 305-307. “Probably the hydrogen bonds between the carbonyl groups with the hydroxyl groups (C=O....HO) as well as between the hydroxyl groups (OH...OH) are responsible for the higher....” but at page 6, lines 239-242. “.....the absence of any shift ofthe peak attributed to the O-H groups, which implies the absence and interaction of this group with Ag, TiO2 and ZnO. So how is it in the end?   Figures 6-13. It must be specified which and how the evidence is. I suggest that a, b, c, d be inserted in each figure.   I consider that this part: "Determination of in vitro antimicrobial activity" looks like a report and not like a part of a scientific paper. I suggest that this part be rewritten.    

At this moment, in the current form, this manuscript cannot be published in Nanomaterials.

Author Response

...The scientific subject could be of interest from practical point of view, yet, the format and the structure of the manuscript is not acceptable.

 Response

Initially we would like to thank the reviewer for the constructive comments that helped improving the quality of our manuscript. A point-by-point response to each comment follows. All changes have been marked with track changes in the revised version.

Please make sure that the title of your manuscript is carefully crafted to optimize visibility and enhance the publication's overall impact. In this form it is too general.

Response

Thank you for the comment. The title has been modified to present better our findings.

Page 1, line 6. Laboratory not Lab.

Response

Changed

The abstract is not a convincing one; it is too short, slightly minded caching… Also, does not quite draw attention of the reader. I think the first sentence (page 1, lines 11-13) should be deleted. Page 1, line 25, where does the abbreviation PEGMEMA come from?

Response

A large part of the abstract has been rewritten and all comments have been taken into account

I did not find anywhere how Ag NPs were obtained. On page 4, line 173 it says that AgNO3 was used.

Response

Thank you for the comment. Extra comments have been added in section 2.3 on how Ag NPs are formed from AgNO3.

Page 5, line 227. Figure 1a-1d?

Response

Changed to Figure 1

Page 6, line 230 “....is located as a strong and broad peak...” But from figure 1 this is not observed.

Response

Changed to broad peak only.

Page 6, line 235. Please add bibliographic references here.

Response

Thank you. It was added, #38

Page 6, line 253. Please add bibliographic references here.

Response

Thank you it was added. #39

Page 8, line 280.” For silver the peaks observed at 38.2o and 44.4o...” From your figure I do not notice the 2 peaks.

Response

Thank you for the comment. Some of the XRD scans were measured again and both Figure 3a and 3b were redrawn; indicative angles are highlighted in order to show better the results. Some comments have been added also in the text.

Page 9, line 297. “...containing ZnO and TiO2 show greater thermal stability” I think that is too much said. Perhaps it is better to add that it has a slightly improved thermal stability compared to the one observed for the PEGMA graded sample.

 Response

The phrase has been changed according to the comment.

Page 9, lines 305-307. “Probably the hydrogen bonds between the carbonyl groups with the hydroxyl groups (C=O....HO) as well as between the hydroxyl groups (OH...OH) are responsible for the higher....” but at page 6, lines 239-242. “.....the absence of any shift ofthe peak attributed to the O-H groups, which implies the absence and interaction of this group with Ag, TiO2 and ZnO. So how is it in the end?

Response

Thank you for the comment. The text has been modified to be consistent with the FTIR findings.

Figures 6-13. It must be specified which and how the evidence is. I suggest that a, b, c, d be inserted in each figure.

Response

Indeed the addition of a, b, c, d in each figure was mandatory and they were added including comments in each figure legend. Thank you for the comment.

I consider that this part: "Determination of in vitro antimicrobial activity" looks like a report and not like a part of a scientific paper. I suggest that this part be rewritten.

Response

Some parts of this section were re-written. Indeed, originally we were thinking about adding the pictures with the antimicrobial activity as a supplementary file, but then we decided to keep them in the text since this is the main interesting part of this study.

Reviewer 2 Report

Comments and Suggestions for Authors

Synthesis, characterization and antimicrobial activity of nanocomposite materials based on poly(ethylene glycol methacrylate)s with silver, TiO2 or ZnO nanoparticles

1.       The abstract section is too prolix. Specific findings and objectives can be highlighted.

2.       There is no peaks related to ZnO and TiO2 seen in the XRD patterns. The XRD patterns can be analyzed again.

3.       The rational design of the Ti, Zn and Ag based metal oxides can be explained with the reference of following articl;es: doi.org/10.1016/j.ijhydene.2020.09.213, doi.org/10.1016/j.inoche.2023.110675

4.       Morphology and microstructural analysis are recommended. TEM and FESEM images should be provided.

5.       The antibacterial activity of the Escherichia Coli (E.coli) on PEGMEMA polymer and PEGMEMA with Ag/NPs, TiO2/NPs and ZnO/NPs is very low why?

Author Response

Synthesis, characterization and antimicrobial activity of nanocomposite materials based on poly(ethylene glycol methacrylate)s with silver, TiO2 or ZnO nanoparticles

Response

Initially we would like to thank the reviewer for the constructive comments that helped improving the quality of our manuscript. A point-by-point response to each comment follows. All changes have been marked with track changes in the revised version.

 The abstract section is too prolix. Specific findings and objectives can be highlighted.

Response

A large part of the abstract has been rewritten and all comments have been taken into account.

  1. There is no peaks related to ZnO and TiO2 seen in the XRD patterns. The XRD patterns can be analyzed again.

Response

Thank you for the comment. Some of the XRD scans were measured again and both Figure 3a and 3b were redrawn; indicative angles are highlighted in order to show better the results. Some comments have been added also in the text.

  1. The rational design of the Ti, Zn and Ag based metal oxides can be explained with the reference of following articles:  doi.org/10.1016/j.ijhydene.2020.09.213doi.org/10.1016/j.inoche.2023.110675

Response

Extra comments have been added in the Introduction and the Antibacterial section and both suggested references were added #27, #28.

  1. Morphology and microstructural analysis are recommended. TEM and FESEM images should be provided.

Response

Indeed adding TEM or FESEM images would be beneficial and would provide some information. However, since the main objective of this work was to examine the antimicrobial activity of the nanoparticles in the polymer matrix and since we do observe antimicrobial activity we consider it an implicit proof that nanoparticles exist in the polymer matrix in homogeneous dispersion. The procedure followed for the synthesis of the nanocomposites (i.e. in situ polymerization) helps also in this direction. The second reason for not including such measurements was that such instruments were not available this time and we would have to wait for more than 1 month for such measurements, which is not allowed.

  1. The antibacterial activity of the Escherichia Coli (E.coli) on PEGMEMA polymer and PEGMEMA with Ag/NPs, TiO2/NPs and ZnO/NPs is very low why?

Response

We are sorry but we do not understand the comment. As shown in Figure 7, and reported in the text the antibacterial activity of PEGMEMA with Ag/NPs, TiO2/NPs and ZnO/NPs (Fig. 7b-d) against E. coli is extremely good, since no colonies were formed compared to the PEGMEMA polymer (Fig. 7a).

Round 2

Reviewer 1 Report

Comments and Suggestions for Authors

Nanomaterials Manuscript Number: nanomaterials-2831177 Title: "Synthesis of novel nanocomposite materials with enhanced antimicrobial activity based on poly(ethylene glycol methacrylate)s with Ag, TiO2 or ZnO nanoparticles" Author(s): Melpomeni Tsakiridou, Ioannis Tsagkalias, Rigini M. Papi, Dimitris S. Achilias                        

The authors have performed the required revision on their manuscript, giving careful point-by point responses, according to reviewer remarks.   The work has been improved, the article entitle " Synthesis of novel nanocomposite materials with enhanced antimicrobial activity based on poly(ethylene glycol methacrylate)s with Ag, TiO2 or ZnO nanoparticles” authored by Melpomeni Tsakiridou, Ioannis Tsagkalias, Rigini M. Papi, Dimitris S. Achilias is now ready for publication in “Nanomaterials”.